# Improving Uptake and Sustainability of Sanitation Interventions in Timor-Leste: A Case Study

**DOI:** 10.3390/ijerph18031013

**Published:** 2021-01-24

**Authors:** Naomi E Clarke, Clare E F Dyer, Salvador Amaral, Garyn Tan, Susana Vaz Nery

**Affiliations:** 1Kirby Institute, University of New South Wales, Sydney, NSW 2052, Australia; nclarke@kirby.unsw.edu.au (N.E.C.); cdyer@kirby.unsw.edu.au (C.E.F.D.); 2Research School of Population Health, College of Health & Medicine, The Australian National University, Canberra, ACT 2601, Australia; salvador.amaral@menzies.edu.au (S.A.); garyntan@gmail.com (G.T.); 3Menzies School of Health Research, Charles Darwin University, John Mathews Building, Royal Darwin Hospital Campus, Tiwi, NT 0810, Australia

**Keywords:** sanitation, WASH, CLTS, ODF, subsidies, Timor-Leste

## Abstract

Open defecation (OD) is still a significant public health challenge worldwide. In Timor-Leste, where an estimated 20% of the population practiced OD in 2017, increasing access and use of improved sanitation facilities is a government priority. Community-led total sanitation (CLTS) has become a popular strategy to end OD since its inception in 2000, but evidence on the uptake of CLTS and related interventions and the long-term sustainability of OD-free (ODF) communities is limited. This study utilized a mixed-methods approach, encompassing quantitative monitoring and evaluation data from water, sanitation, and hygiene (WASH) agencies, and semi-structured interviews with staff working for these organizations and the government Department of Environmental Health, to examine sanitation interventions in Timor-Leste. Recommendations from WASH practitioners on how sanitation strategies can be optimized to ensure ODF sustainability are presented. Whilst uptake of interventions is generally good in Timor-Leste, lack of consistent monitoring and evaluation following intervention delivery may contribute to the observed slippage back to OD practices. Stakeholder views suggest that long-term support and monitoring after ODF certification are needed to sustain ODF communities.

## 1. Introduction

Access to clean water and adequate sanitation was classified in 2010 by the United Nations as a human right, and improving equity of access to sanitation represents a key development goal [1]. The Millennium Development Goal target for sanitation—to halve the proportion of the population without sustainable access to sanitation by 2015—was missed by over 700 million people [2]. The updated Sustainable Development Goals (SDGs) include the ambitious targets of achieving access to equitable sanitation for all and ending open defecation by 2030 [3]. However, the 2017 Joint Monitoring Programme report found that no SDG region, except Australia and New Zealand, is on target to reach this goal [4]. In 2017 an estimated 673 million people still practiced open defecation worldwide [5].

It is widely accepted that access to adequate sanitation is associated with improved health outcomes. Sanitation access has been shown to significantly decrease the risk of diarrheal disease [6,7]. In 2017, unsafe sanitation was estimated to cause 774,000 deaths worldwide, of which 533,768 were children under 5 years old [8,9]. The global disease burden of unsafe sanitation was estimated at 41.5 million disability-adjusted life years, with the majority of this burden being due to diarrheal disease [9,10].

However, several recent trials examining the impact of large-scale sanitation interventions on health outcomes have failed to detect impacts on outcomes, including diarrhea incidence and soil-transmitted helminth infections [11,12,13,14,15,16]. In many cases this was attributed to the low uptake of sanitation interventions. Indeed, achieving and sustaining high latrine coverage and usage has proved a substantial challenge for the water, sanitation, and hygiene (WASH) sector. Despite significant investments in sanitation by national governments and international and national non-governmental organizations (NGOs), results have often fallen short of expectations [17].

There is considerable debate surrounding the optimal way to improve the coverage and use of improved sanitation facilities. “Supply-led” sanitation programs that rely heavily on subsidies—i.e., those that provide sanitation hardware or funds to build latrines—have been criticized for failing to achieve community involvement or ownership of latrines, resulting in poor uptake and use of latrines and community dependence on subsidies [18,19,20]. In response to these concerns, “demand-led” interventions to improve sanitation have gained popularity. Community-led total sanitation (CLTS), pioneered in 2000 in Bangladesh, represents a “zero subsidy” approach that is based on triggering communities into social mobilization, and empowering community members to construct household latrines in order for communities to achieve “open defecation free” (ODF) status [21].

CLTS is now used in more than 60 countries worldwide [22]. However, the CLTS approach has also attracted criticism; there are concerns that it could be viewed as coercive, given its strong emphasis on shame and social stigma [23,24,25]. Furthermore, the capacity for extremely poor communities to construct high-quality, durable latrines may be limited in the absence of any subsidies, leading to a risk of “slippage” (reversion to open defecation) [26,27,28]. Furthermore, monitoring and follow-up occurs in only a small proportion of CLTS programs [29].

Empirical evidence regarding the uptake and sustainability of CLTS and other approaches to improving sanitation is limited [30,31]. Only one randomized controlled trial (RCT) has directly compared the uptake rates of various strategies; this study compared a subsidy-based approach, a community motivation approach based on CLTS, and a market access approach designed to increase availability of materials [32]. The study concluded that only the subsidy-based approach led to a significant increase in hygienic latrine ownership [32]. A meta-analysis examining latrine coverage and use following sanitation interventions found that all types of sanitation interventions resulted in modest increases in latrine coverage, with an average increase of just 14% [30].

Few peer-reviewed studies have examined the opinions and perspectives of the implementers of sanitation programs, and existing studies have mainly focused on CLTS programs. For example, Sigler and colleagues examined the behavioral change frameworks used by implementers of CLTS programs and explored which activities these implementers considered most valuable [29]. Similarly, Lawrence and colleagues examined the perspectives of CLTS stakeholders, including barriers to latrine construction [33]. On the other hand, we identified no published literature examining stakeholder opinions on subsidized sanitation.

The current study was conducted in Timor-Leste, where access to improved sanitation remains poor, especially in rural areas. In 2015, an estimated 70% of rural households did not have access to improved sanitation [34]. The country’s WASH program is led by the Ministry of Public Works, under the Direcção de Agua e Saneamento (Directorate of Water and Sanitation), which is supported by several other ministries, NGOs, and international agencies. The government has prioritized a decentralized approach since 2016, with WASH project implementation being led by international and bilateral agencies working with local NGOs at the municipality level. The national sanitation policy in Timor-Leste states that, “The construction of household toilets and other household sanitation facilities shall not be subsidized except in specific situations where the households are disadvantaged” [35]. The majority of NGOs in Timor-Leste utilize a modified version of CLTS called “PAKSI” (Planu Asaun Komunidade Saneamentu no Ijiene/Participatory Community Action to Sanitation and Hygiene) which focusses on disgust rather than shame as the main trigger for behavioral change [36]. One organization provides households with a hardware subsidy to facilitate latrine construction. Households are required to dig their own pit and construct their own latrine, with assistance from the organization. The hardware subsidy includes cement, a plastic pour/flush pan, and other construction materials.

In 2015, a study in Timor-Leste investigated the status of 45 communities that had been certified as ODF at least two years previously, following CLTS interventions, and found that 14 out of 45 communities (31%) retained ODF status [37]. Approximately 20% of households had reverted to open defecation, and 68% of households had their own latrines. Key motivators for sustaining ODF were health improvements and emotional factors such as shame, disgust, pride, and fear. Barriers included competing household priorities and poor quality of initial latrine construction [37].

In this study, conducted in 2017–2018, we aimed to complement this existing knowledge by examining monitoring data from sanitation interventions within Timor-Leste and exploring the opinions of sanitation program implementers and stakeholders of both subsidized approaches and PAKSI/CLTS.

## 2. Methods

### 2.1. Ethics Approval and Consent to Participate

This study obtained ethical approval from the Human Research Ethics Committee at the Australian National University (2017/489). Informed written consent was obtained from all study participants.

### 2.2. Study Design and Data Collection

This mixed-methods study employed two approaches to understand the uptake and sustainability of sanitation interventions in Timor-Leste. Monitoring and evaluation data collection and stakeholder interviews were undertaken between September 2017 and January 2018.

Firstly, six international organizations implementing WASH projects in Timor-Leste were contacted and asked to provide monitoring records for sucos (villages) in which sanitation interventions had been implemented in the preceding five years (2012–2017). Specifically, information was requested on: project name and funding source; names of partner NGO(s); municipality and suco names; number of households; dates of commencement and completion of interventions; monitoring date(s); the number, type, and usage of household latrines at any timepoint since the intervention; ODF status declaration date (if applicable); and ODF status at any timepoint since the intervention.

Secondly, stakeholder interviews were carried out with staff members working for twelve organizations that implement WASH interventions, including the six international organizations and their local partners, to understand their views and experiences. All major organizations working on WASH interventions projects in Timor-Leste were contacted (including international agencies, local NGOs, and bilateral organizations), and the Department of Environmental Health at the Timor-Leste Ministry of Health. Each organization was asked to provide a list of all employees managing WASH projects; all individuals were contacted for interview. Participants included WASH field staff, WASH program managers, and government officials.

Interviews were semi-structured and were conducted either face-to-face or over the phone, with a mixture of multiple-choice questions (with answers read out by the interviewer), opinion statements, and an opportunity for free responses. Multiple-choice questions explored stakeholders’ opinions of the main barriers to achieving and maintaining ODF status, as well the main reasons communities achieve ODF. Stakeholders were also asked about the main advantages and disadvantages of CLTS and subsidies. Stakeholders were asked to what extent they agreed or disagreed with statements relating to achieving and maintaining ODF status, and how they thought current sanitation intervention strategies could be improved. WASH organization leaders were not informed as to who had participated in the research project to maintain confidentiality, and to ensure that participants felt able to respond honestly to the questions.

All interviews were conducted by the same interviewer in Tetum or English in a private setting. Interviews were not recorded, and answers were entered directly into a password-protected Microsoft Access database.

### 2.3. Data Analysis

Monitoring and evaluation data received were processed and collated in an Excel spreadsheet. Information was extracted from the spreadsheet on the number of communities that received sanitation interventions, and number of communities certified as ODF, as a measure of uptake of sanitation interventions. Other information, such as sanitation indicators reported by each agency, timing and frequency of follow-up monitoring, level of monitoring (household vs. community), and whether projects were monitored after ODF certification, was also summarized.

Responses to the interviews were exported from Microsoft Access into Stata 14.1 (Stata Corporation, College Station, TX, USA) for analysis. Proportions and their 95% confidence intervals were calculated for each question response.

Qualitative responses were analyzed to identify key themes and recommendations.

## 3. Results

### 3.1. Uptake and Sustainability of Sanitation Interventions

Data were received from five out of six international WASH agencies responsible for implementing water and sanitation projects in Timor-Leste. Four agencies used interventions based on CLTS techniques, and one used a “partial subsidies” approach.

Table 1 summarizes the type of monitoring and evaluation data provided by each agency. One agency provided only a summary of their projects stating the number of communities triggered (but not triggering dates), and number of communities certified as ODF; the other four agencies provided more detailed information about household latrine coverage for individual WASH projects. Four agencies provided data on the proportion of communities certified as ODF. Of these, three agencies provided data for individual projects, and one provided aggregate data only.

Latrine coverage and ODF status for projects implemented between 2012 and 2017 were the main sanitation indicators reported in the monitoring data. Data from over 400 sucos across all thirteen municipalities suggest moderate success in achieving ODF status. The average proportion of ODF communities after the CLTS-inspired intervention ranged from 33% to 75% across the different agencies, and average community-level household latrine coverage ranged from 23% to 99% across the different agencies at the time of the last available monitoring, which was usually at the project end.

Ongoing monitoring following project completion was rarely documented, with only two agencies measuring latrine coverage and ODF status after project completion. One agency carried out monitoring every 3 months between March and December 2015 for projects completed since January 2012. Average latrine coverage across all sucos at each timepoint was between 64% and 70%, with 84% of households overall having a latrine at the end of the monitoring in December 2015 (data not shown). The agency using a partial-subsidy approach had also conducted a “lookback” study 2–6 years after completion of nine WASH projects across seven out of thirteen municipalities, and chosen as a representative sample, to understand slippage rates. The study found 86% of latrines were still functional, with 14% of households reverting back to OD (data not shown) [38].

The other three agencies provided monitoring information at project completion only, with no follow-up data provided. Both triggering dates and project completion dates ranged from 2007 to 2016.

### 3.2. Stakeholder Interviews

A total of thirty-one interviews were conducted with staff members of six international agencies (twenty interviews), six local NGOs (nine interviews), and the Timor-Leste Ministry of Health (two interviews) between September 2017 and January 2018. Fourteen interviews were conducted face-to-face, two were completed electronically due to time constraints, and fifteen were conducted over the telephone.

Table 2 shows stakeholders’ opinions on enablers and barriers to achieving and maintaining ODF status. Opinions on different approaches to achieving ODF status are summarized in Table 3.

Stakeholders indicated that, in their experience, the two main reasons communities achieve ODF status are a belief that it will improve health (87.1%, 95% CI 68.9–95.4), and that latrines offer increased privacy (83.9%, 95% CI 65.4–93.5). In addition, 71.0% (95% CI 51.8–84.8) of stakeholders cited shame or disgust of OD as a main enabler, whereas financial incentives were only considered a key factor by 16.1% (95% CI 6.5–34.6).

Stakeholder opinions were more divided on the main barriers to achieving and maintaining ODF status, with a broad range of views evident. The majority of stakeholders believed that lack of community leadership was an important barrier to achieving ODF status (64.5%, 95% CI 45.5–79.9), along with communities believing that government rather than individuals should pay for improved facilities (61.3%, 95% CI 42.4–77.3), and an unwillingness within the community to change routines or behaviors (61.3%, 95% CI 42.4–77.3).

The majority of stakeholders indicated that in their experience, lack of follow-up by local government authorities was a major barrier to maintaining ODF status (61.3%, 95% CI 42.4–77.3). Almost half of all stakeholders cited factors relating to poor latrine quality as a major barrier, such as poor-quality latrines that fill up quickly or break easily (48.4%, 95% CI 30.8–66.4), a lack of materials to build or improve latrines (48.4%, 95% CI 30.8–66.4), and households being unable to afford to build or maintain latrines (48.4%, 95% CI 30.8–66.4).

The majority of stakeholders believed that CLTS/PAKSI interventions were the most effective in achieving permanent ODF status (71.0%, 95% CI 51.8–84.8), with no stakeholder indicating that subsidies (material or financial) alone were effective.

The majority of stakeholders cited the low cost of CLTS (83.8%, 95% CI 65.4–93.4), and its empowerment of local community leaders (77.4%, 95% CI 58.4–89.3) as advantages. The main disadvantages were less clear-cut, with just over half of the stakeholders identifying poor-quality latrines (54.8%, 95% CI 36.4–72.0), and the same number indicating challenges in finding suitable local facilitators (54.8%, 95% CI 36.4–72.0). Approximately half of the stakeholders indicated that communities tended to revert to past behaviors (51.6%, 95% CI 33.6–69.2), though it is unclear if stakeholders are directly attributing this reversion in behavior to the CLTS intervention.

Nearly two thirds of stakeholders indicated that financial subsidies allow construction of better-quality latrines (64.5%, 95% CI 45.5–79.9), and ultimately improve chances of achieving ODF status (54.8%, 95% CI 36.4–72.0). Disadvantages of financial subsidies were cited almost equally among stakeholders, with over half of respondents citing the cost of such subsidies to organizations (58.1%, 95% CI 39.4–74.7), and that subsidies reduce community involvement in addressing OD (58.1%, 95% CI 39.4–74.7). Stakeholders reported that whilst subsidies may result in more latrines being constructed, communities may be less likely will use these latrines (58.1%, 95% CI 39.4–74.7).

Appendix A shows level of stakeholder agreement with statements relating to the roll-out of interventions in communities. Over 80% of stakeholders agreed or strongly agreed that, “Encouraging feelings of shame/disgust is an effective way of achieving and maintaining ODF status,” and over 60% of respondents disagreed or strongly disagreed that, “Simple pit latrines aren’t effective in maintaining ODF status.” The statement drawing the greatest division of responses was that, “It is more difficult to achieve ODF status with CLTS strategies when latrines are subsidized,” with 39% of stakeholders agreeing or strongly agreeing, and 42% of stakeholders disagreeing or strongly disagreeing.

All 31 participants provided suggestions for how to improve current sanitation intervention strategies. Five key themes were identified, relating to (1) cooperation, collaboration, and leadership; (2) behavior change; (3) subsidies; (4) follow-up and monitoring; and (5) water. Table 4 provides selected representative quotes relating to each of these five themes from interview responses.

Forty-eight responses across twenty-five participants mentioned concepts of leadership, collaboration, and cooperation, ranging from government, to local authority, to local leadership level. A common subtheme related local leadership being crucial to achieve and maintain ODF status, with local laws or policies to back up local leaders suggested. 

The theme of behavior change, often mediated through health promotion and education, was represented in forty-six responses from twenty-seven participants, and often overlapped with the theme of leadership. Behavior change is at the heart of CLTS-based interventions, and responses suggested that continued health promotion and education is needed to reinforce messages communicated during the initial triggering process to sustain ODF. Suggestions included WASH promotion, increasing knowledge and awareness around sanitation and hygiene, and ensuring sanitation policies are explained clearly to communities. Ongoing education around how to build and maintain latrines, and how to use available subsidies were also suggested.

Forty-three responses from twenty-four participants discussed the role of subsidies in achieving and maintaining ODF status. Opinions were divided between those who thought subsidies had a role in achieving and maintaining ODF status, and those who felt subsidies should be avoided. Eleven responses from those who favored subsidies noted that they should be targeted towards the vulnerable or elderly; three respondents also suggested subsidies should be reserved only for communities that have already achieved ODF status (for example to help upgrade or maintain latrines). Reasons given for avoiding subsidies are similar to those used to rationalize the CLTS approach, such as subsidies being used for other purposes, and reducing proactivity in communities.

Considerable overlap was also noted with the theme of leadership and the need for ongoing monitoring and support after ODF status is achieved. Twelve responses noted the need for ongoing monitoring after project completion by governments, local governments, NGOs, and local leaders, to reinforce and sustain the sanitation campaign and ODF status. One respondent summarized the importance of ongoing monitoring: “If there is no control and follow up there will be no success and there will be no ownership.”

## 4. Discussion

The results of this study show that CLTS-inspired approaches to eliminating OD in Timor-Leste have mostly been successful initially, with monitoring data showing increases in latrine coverage at completion of interventions. However, ongoing monitoring after project completion is rarely conducted, and the long-term sustainability of these interventions is therefore unclear.

We have presented a range of views from stakeholders involved in WASH interventions and sanitation projects in Timor-Leste. Opinions were diverse, but recurrent themes highlight the need for long-term support and monitoring after ODF certification to sustain ODF communities.

It was difficult to draw strong conclusions on uptake and sustainability of CLTS and related interventions from the monitoring and evaluation information provided. There was considerable variability in the indicators reported, the quality of data, and mechanisms of data collection. This variability indicates a need for agreement on key indicators to be measured, and frequency and timing of reporting across WASH organizations. 

In order to contextualize stakeholder interview findings, we compared and contrasted these with community perspectives presented in the Partnership for Human Development’s Report on ODF sustainability in Timor-Leste (ODF-SR), which presented the views of community members from 290 households across eighteen aldeias (hamlets) on similar topics [37].

Stakeholders in our study felt that improvements in health, and shame or disgust of OD were the two main reasons communities achieve and maintain ODF status, which mirror those reported by community members in the ODF-SR [37]. In a broader ODF sustainability report carried out by Plan International in four countries in Africa, these two factors were also identified by community members as the main motivators for achieving and maintaining ODF status [27]. These findings are consistent with the rationale underpinning the CLTS approach [22].

Quality and durability of latrines were also cited by stakeholders and community members as important motivators for maintaining ODF status [37]. However, a majority of stakeholders believed that a simple pit latrine was sufficient to achieve ODF status, which points to the difference in long- and short-term objectives of such projects. Pit latrines may facilitate rapid ODF certification, and perceived project success, but often fall into disrepair, and households may have little incentive, time, or resources to repair them [39].

Stakeholders frequently mentioned lack of long-term follow-up, monitoring, or support following ODF certification, either by NGOs or local government officials. With SDG indicators setting targets for equitable sanitation and hygiene for all, in countries where OD is a health challenge, governments have focused on achieving ODF status in communities, with little consideration for ODF sustainability [40]. This has resulted in reversion to OD, as reported in the ODF-SR, because follow-up support, health promotion, campaigning, and financial assistance are lacking [37]. Funding for CLTS and other sanitation interventions are usually external, and budget and capacity rarely extend beyond the delivery of the intervention, with minimal funding for ongoing monitoring [41]. Long-term government engagement and commitment, in particular allocating budget and capacity for follow-up visits to ODF communities, is critical to long-term sustainability of ODF [39,40,42]. Such involvement by government may incentivize communities to remain ODF and empower local leaders [39,40,42].

Stakeholders frequently suggested that health promotion and education activities should continue long-term to sustain behavior change and ODF practice. Health promotion and education are core elements of the CLTS approach but need to be reinforced after the project is complete. Studies have shown that health promotion activities are crucial to sustain behavioral change and ODF practice [43,44]. Furthermore, since community achievement of ODF status is only category two out of five in Timor-Leste’s National Basic Sanitation Policy, ongoing promotion and education activities are vital for communities to progress to category 3 (hygienic), category 4 (solid-waste free), and finally category 5 (foul water free) of the framework [35].

Most stakeholders agreed that financial subsidies allow construction of better-quality latrines, and improve chances of achieving ODF status. Many responses suggested financial incentives are required but only using a targeted approach, and to enable communities that are already declared ODF to improve their latrines. Provision of materials, rather than monetary subsidies, was suggested to avoid frequently-cited challenges of “subsidies will create laziness” or households using financial subsidies “for other purposes.” Given that latrine quality is strongly linked to ODF maintenance, financial or material support to enable latrine improvement will likely have a strong impact on whether communities are able to remain ODF [27,42].

In the ODF-SR, community members considered water essential to both toilet use and hygiene, and thus sustainability of ODF [37]. Household water access enables construction of better-quality toilets (such as pour-flush latrines) and increases chances of sustaining ODF status. Improved toilets are less likely to break, require less maintenance, and confer a greater sense of pride and desire to use them [27,30,37]. However, whilst water access was mentioned in ten stakeholder responses, only one stakeholder made the link between water and latrine quality. Improving water infrastructure in parallel with delivering CLTS and other sanitation interventions is vital for the long-term sustainability of ODF status [37].

Several stakeholders mentioned the need for laws or legislation to support local leaders in their role in maintaining ODF status in the community; interview findings also show that 87% of stakeholders agreed or strongly agreed that “encouraging feelings of shame or disgust is an effective way of achieving and maintaining ODF status.” However, the poorest and most vulnerable members of communities may have neither the financial assets nor physical ability to build their own latrine, and this can leave them feeling ostracized and vulnerable to stigma from other community members, and at risk of fines or shaming [23,45]. Indeed, this argument underlies why one of the five agencies provides partial subsidies to help communities construct latrines [38]. CLTS is based on a principle of an “idealized notion of community,” but inequalities, conflict, and patronage can act as barriers to CLTS implementation [46]. CLTS methods to create social pressure and OD behavioral change may infringe on human rights, such as an individual’s right to dignity [23,24]. Therefore, laws or policies to facilitate maintenance of ODF practices must be considered through the lens of equity and human rights—is everyone within the community financially and physically able to construct and maintain a latrine, and if not, what measures can and should be taken to facilitate this?

### Limitations of Study

There are several limitations to this study. The study was based in one country, and interviews were conducted with individuals working for a limited number of organizations. Therefore, experiences in other countries may be different to those presented here, and findings may not be generalizable. Due to the small numbers of stakeholders available to interview, and confidentiality challenges, we could not stratify our analysis by level of staff responsibility, type of sanitation implementation, or organization type. Data were collected at the end of 2017, and so the context, organizational priorities, and monitoring processes may have changed since then. However, as of April 2019, only four out of thirteen municipalities in Timor-Leste were declared ODF, suggesting the findings of this study are still relevant, especially given Timor-Leste’s commitment to achieve countrywide ODF status by 2020 [47].

Interviews were conducted on the telephone, face-to-face, or completed online, and this variation may have biased the responses stakeholders gave. For example, individuals may have been more inhibited in a face-to-face interview vs. a telephone interview. In cases where stakeholder views differed from their organization’s principles or guidelines, this may have particularly impacted how comfortable they felt providing honest answers.

Finally, whilst we included a mix of experience levels and roles for stakeholders, the study had a relatively small sample size, and we only report here impressions from a few stakeholders and communities. Whilst we have used the ODF-SR as our primary comparator when considering different viewpoints and opinions, both our report and the ODF-SR only sampled small numbers of individuals, and there are likely still gaps in our knowledge.

## 5. Conclusions

This study used Timor-Leste as a case study to understand the uptake and sustainability of sanitation interventions, and the perspectives and opinions of stakeholders working in the WASH sector. From analysis of monitoring and evaluation data from WASH agencies in Timor-Leste, and stakeholder interviews, recommendations were made to improve sustainability of sanitation interventions. These study findings indicate firstly that targeted material subsidies may improve the long-term sustainability of ODF communities; secondly, that collaboration is needed between NGOs and local government to develop consistent processes for monitoring and evaluation both during project implementation, and after project completion; and thirdly, that long-term health promotion and community support activities are needed after project completion to facilitate sustainable behavioral changes. Whilst the Timor-Leste government has prioritized a decentralized approach to WASH projects, these study findings suggest that government oversight and involvement may be needed to ensure a unified and coordinated approach across all international and national WASH agencies working in the country. Continuing to investigate and refine approaches to improving sanitation is essential to ensure sustainable changes and long-term benefits to communities.

## Figures and Tables

**Table 1 ijerph-18-01013-t001:** Summary of monitoring and evaluation data submitted by each water, sanitation, and hygiene (WASH) agency.

Organization	Intervention Approach	Monitoring after Project End?	Triggering Date Provided	ODF Status Reported	Latrine Coverage Reported	Reporting Level
1	CLTS-inspired	✓	✓	✓	✓	Village level
2	CLTS-inspired	×	×	✓	✓ ^†^	Village level
3	CLTS-inspired	×	✓	✓	✓	Village level
4	Partial subsidy	✓	×	×	✓	Village level *
5	CLTS-inspired	?	×	✓	×	Aggregate

* Sub-village level detail also provided; ^†^ latrine usage also reported. ODF: Open defecation free; ✓: Yes; ×: No; ?: Unclear from information provided.

**Table 2 ijerph-18-01013-t002:** Stakeholder views (*n* = 31) on factors relating to achieving and maintaining open defecation free (ODF) status in the community.

Factors	*n*	% (95% CI)
Main reasons communities achieve ODF status		
Community members know it will improve health	27	87.1% (68.9–95.4)
Latrines offer increased privacy	26	83.9% (65.4–93.5)
Community members are ashamed/disgusted by OD	22	71.0% (51.8–84.8)
Social pressure from other community members	22	71.0% (51.8–84.8)
Sense of civic responsibility to have a latrine	20	64.5% (45.5–79.9)
Latrines are more convenient	18	58.1% (39.4–74.7)
Offer of financial incentives	5	16.1% (6.5–34.6)
Main barrier to achieving ODF status in a community		
Lack of community leadership	20	64.5% (45.5–79.9)
Belief government should pay for improved facilities	19	61.3% (42.4–77.3)
Unwillingness to change routine or behavior	19	61.3% (42.4–77.3)
Improved sanitation not a priority	14	45.2% (28.0–65.3)
Cannot afford to buy or build latrines	10	32.3% (17.7–51.4)
Lack of trust in NGO workers	9	29.0% (15.2–48.2)
Main barrier to maintaining ODF status in a community		
Lack of follow-up by local government authorities	19	61.3% (42.4–77.3)
Unaffordable to build new latrines/maintain existing latrines	16	51.6% (33.6–69.2)
Lack of materials for latrine construction and/or improvement	15	48.4% (30.8–66.4)
Poor quality latrines: break frequently/pits fill quickly	15	48.4% (30.8–66.4)
Latrine maintenance “too much work”	9	29.0% (15.2–48.2)
Lack of follow-up by NGOs	6	19.4% (8.5–38.1)

NGO: Non-governmental organization; ODF: Open defecation free; OD: Open defecation.

**Table 3 ijerph-18-01013-t003:** Stakeholder views (*n* = 31) on sanitation interventions used to achieve ODF status in the community.

Factors	*n*	% (95% CI)
Most effective intervention to achieve *permanent* ODF status		
CLTS/PAKSI	22	71.0% (51.8–84.8)
Combination of CLTS and targeted subsidies	6	19.4% (8.5–38.1)
Providing subsidies in the form of materials to build latrines	1	3.2% (0.4–21.6%)
Providing financial subsidies	0	0% (N/A)
The main advantages of CLTS		
Relatively inexpensive	26	83.8% (65.4–93.4)
Encourages growth of local natural leaders	24	77.4% (58.4–89.3)
Leads to long-term change by modifying behavior and attitudes	20	64.5% (45.5–79.9)
Does not rely on subsidies or service delivery from external agents	17	54.8% (36.4–72.0)
People more likely to value/maintain latrines they have paid for	15	48.4% (30.8–66.4)
The main disadvantages of CLTS		
CLTS often results in poor quality latrines	17	54.8% (36.4–72.0)
Finding the right facilitators to implement CLTS is difficult	17	54.8% (36.4–72.0)
Community members commonly revert to past behavior/habits	16	51.6% (33.6–69.2)
Difficult to motivate villagers to change their own behavior	10	32.3% (17.7–51.4)
Not everyone can afford to build/purchase latrines	10	32.3% (17.7–51.4)
Triggering feelings of shame/disgust can lead to tension/conflict	9	29.0% (15.2–48.2)
I don’t know/no opinion	1	3.2% (0.4–21.6%)
The main advantages of providing subsidies		
Usually lead to the construction/purchase of better-quality latrines	20	64.5% (45.5–79.9)
Improve chances of achieving ODF	17	54.8% (36.4–72.0)
Allow everyone the ability to build latrines	15	48.4% (30.8–66.4)
Ensure adequate supply of latrines to meet demand	13	41.9% (25.3–60.6)
Useful in convincing people to change their behavior	13	41.9% (25.3–60.6)
I don’t know/no opinion	3	9.7% (2.9–27.4)
The main disadvantages of providing subsidies		
Communities become reliant on subsidies to build/buy latrines	22	71.0% (51.8–84.8)
Subsidies alone do not lead to long-lasting behavioral change	20	64.5% (45.5–79.9)
Costly and require ongoing funding	18	58.1% (39.4–74.7)
Do not require community participation or encourage leadership	18	58.1% (39.4–74.7)
Increased latrine coverage, but not necessarily increased latrine use	17	54.8% (36.4–72.0)

CLTS: Community-led total sanitation; PAKSI: Planu Asaun Komunidade Saneamentu no Ijiene (Participatory Community Action to Sanitation and Hygiene); ODF: Open defecation free.

**Table 4 ijerph-18-01013-t004:** Selected quotes from stakeholders relating to five main themes.

Selected Quotes		Job Role
Theme 1: Cooperation, collaboration, and leadership (48 responses)		
1	*“[There is a] lack of active government leadership in pushing for ODF”*	National WASH Manager
2	*“Local leader is very important to involve in raising awareness in community”*	WASH Integration Specialist
3	*“Need cooperation between NGO, central government and local leader to encourage community to actively participate in the sanitation project activities to achieve ODF”*	WASH Project Facilitator
4	*“Need continued support from the local authority to encourage communities to maintain the sanitation strategies and sanitation policy”*	WASH Project Manager
Theme 2: Behavior change, education and health promotion (46 responses)		
5	*“$3 Project that was implementing from when the beginning of Timor-Leste independence in 2000 has been destroying the willingness of community participation.”*	WASH Officer
6	*“Need to have continuous health promotion and education in the community by collaborating with the health department and other relevant institutions” **	Sanitation Officer
7	*“Provide training on how to build the proper latrine”*	Area Manager
8	*“Give health promotion to raise the awareness and knowledge on the importance of the latrine”*	Sanitation Officer
Theme 3: Subsidies (43 responses)		
9	*“Provide subsidies to the villages that have already achieve ODF”*	WASH Coordinator
10	*“When subsidies are selected by transparent, open and consistent processes… this enables the poorest of the poor to [build] durable latrines to sustain their practice”*	Program Director
11	*“If subsidies are expected, communities may tend to wait until a subsidy is offered”*	WASH Advisor
12	*“Sometimes after having subsidies the community does not build their latrine”*	Sanitation Officer
Theme 4: Follow-up and monitoring after project completion (12 responses)		
13	*“Monitoring has to be effective to maintain ODF status”*	WASH Manager
14	*“There are no proper triggering and monitoring activities. The triggering process is not well facilitated, and even if it is, then monitoring is not happening well (too long after the triggering process or not happening)”*	Program Director
15	*“Need to have regulations in place to secure the ODF status. Government needs to be continuously monitoring, and other relevant institution need to enforce sanitation campaigns to secure ODF. Need a task force and to involve various institutions in the district”*	NGO Director
Theme 5: Water supply (10 responses)		
16	*“Water supply is key to maintaining sanitation, attitude and behavior change”*	Team Leader
17	*“Availability of water often dictates the type of latrine investment households are willing to make. If there is available water, investments in more permanent facilities will be considered more of a household priority”*	WASH Advisor
18	*“There is lack of access to water”*	CLTS Coordinator

* An example of a response that overlaps theme 1 and 2. CLTS: Community-led total sanitation; NGO: Non-governmental organization; ODF: Open defecation free; WASH: Water, sanitation and hygiene.

## Data Availability

Restrictions apply to the availability of these data. Data were obtained from external organizations and are available from the authors with the permission of the external organizations.

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
