# Peer review of "Improving Uptake and Sustainability of Sanitation Interventions in Timor-Leste: A Case Study"

_ijerph, 2021, doi:10.3390/ijerph18031013_

Round 1
Reviewer 1 Report
The article addresses the challenges that are encountered by stakeholders, community leaders and members, and public health/NGO practitioners who work on OD projects. The contribution is just exceptional. The data project is muti-faceted and takes into account different perspectives. It provides a very clear explanation as to why certain ODF projects fail and why some have success.
Overall, the only concern I have is that there be some editing down of text - it's a little too verbose in the discussion and results. More specifically, Line 163-164 has an error: "One a agency" — change to agency. Also, line 163 just reads awkwardly because I don’t know what “Error! Reference source not found.” means? Was the source disconnected for the monitoring system?
In Line 163 - authors state the this phrase summarizes the information provided by each site. Can you define what it is summarizing in a little more detail. From my point of view “no finding is still a finding…” but you need to define what is not being found (or the how that has come to be the case). I’m not sure that is clearly defined or explained. It is written as if it should be intuited and I’m not sure that the phrase is easily intuited given how often it is addressed throughout the analysis.
Similar issue from 197-201.
Other than those two items (overall edit and clarifying the "Error" message), I consider this manuscript to be a great contribution to our scholarship and understanding of implementing ODF programs.
Reviewer 2 Report
The manuscript addresses a pertinent research problem that contributes important insights to the literature on environmental research and public health. The study area is Timor-Leste and focusses on the successes and challenges associated with open defecation malpractices in certain rural areas. It is important to review progress made by existing interventions. Hence, the study adopted a mixed method research design and used both telephone interviews and face-to-face data collection and some of the monitoring data that were collected by NGOs and related organizations. Thus, data is abstracted from a variety of sources and is triangulated quantitatively and qualitatively. The study has yielded very important results and insights that shed light on the successes made and remaining challenges still to be circumvented in the study area. The discussion of the results clearly depicts their significance and their implications for improved sanitation in the study area. Furthermore, the limitations of the study are also discussed very well although given the extent of their length, they warrant a dedicated section called 'Limitations of the Study'. A major weakness in this paper lie in the Conclusion section which is focusing more on the recommendations that flow from the results instead of also briefly alluding to actual conclusions. Thus there needs to be a brief summary of conclusions and this must be followed by recommendations. Lastly, going through the manuscript I came across few instances in which sentences were either not completed adequately or there were some interruptions presenting as 'ERRORS' in bold. Where sentences are too long they must be subdivided into 2-3 short sentences. In making this comment, I am aware that long sentences are acceptable but they must be reduced to a minimum where possible. Kindly go through my list of corrections so that they can be addressed.
